# Effective Prognostic Model for Therapy Response Prediction in Acute Myeloid Leukemia Patients

**DOI:** 10.3390/jpm13081234

**Published:** 2023-08-07

**Authors:** Maria A. Kolesnikova, Aleksandra V. Sen’kova, Tatiana I. Pospelova, Marina A. Zenkova

**Affiliations:** 1City Hematology Center, Novosibirsk 630051, Russia; marija.com.ka@mail.ru; 2Institute of Chemical Biology and Fundamental Medicine SB RAS, Novosibirsk 630090, Russia; marzen@niboch.nsc.ru; 3Department of Therapy, Hematology and Transfusiology, Novosibirsk State Medical University, Novosibirsk 630091, Russia; post_gem@mail.ru

**Keywords:** acute myeloid leukemia, risk stratification, drug resistance, therapy response

## Abstract

**Simple Summary:**

Acute myeloid leukemia (AML) is a hematopoietic disorder characterized by the malignant transformation and abnormal proliferation of bone marrow-derived myeloid progenitors. To select the optimal treatment regimens and predict the therapy response in AML patients, stratification into risk groups based mostly on genetic factors is carried out. Despite this contemporary approach, tumor cell resistance to chemotherapeutic drugs represents one of the main obstacles for improving survival outcomes in AML. In the present study, a new prognostic scale for risk stratification of AML patients based on the drug responsiveness of tumor cells detected in vitro as well as *MDR1* mRNA/P-glycoprotein expression, tumor origin (primary or secondary), cytogenetic abnormalities, and aberrant immunophenotype was developed. Using correlation, ROC, and Cox regression analyses, we demonstrated that the risk stratification of AML patients in accordance with the developed prognostic scale correlates well with the response to therapy and represents an independent predictive factor for the overall survival of patients with newly diagnosed AML.

**Abstract:**

Acute myeloid leukemia (AML) is a hematopoietic disorder characterized by the malignant transformation of bone marrow-derived myeloid progenitor cells with extremely short survival. To select the optimal treatment options and predict the response to therapy, the stratification of AML patients into risk groups based on genetic factors along with clinical characteristics is carried out. Despite this thorough approach, the therapy response and disease outcome for a particular patient with AML depends on several patient- and tumor-associated factors. Among these, tumor cell resistance to chemotherapeutic agents represents one of the main obstacles for improving survival outcomes in AML patients. In our study, a new prognostic scale for the risk stratification of AML patients based on the detection of the sensitivity or resistance of tumor cells to chemotherapeutic drugs in vitro as well as *MDR1* mRNA/P-glycoprotein expression, tumor origin (primary or secondary), cytogenetic abnormalities, and aberrant immunophenotype was developed. This study included 53 patients diagnosed with AML. Patients who received intensive or non-intensive induction therapy were analyzed separately. Using correlation, ROC, and Cox regression analyses, we show that the risk stratification of AML patients in accordance with the developed prognostic scale correlates well with the response to therapy and represents an independent predictive factor for the overall survival of patients with newly diagnosed AML.

## 1. Introduction

Acute myeloid leukemia (AML) is a clonal hematopoietic disorder characterized by the malignant transformation and abnormal proliferation of bone marrow-derived, self-renewing stem cells, or CD34+ CD38− myeloid progenitors [1,2,3]. AML is the most common acute leukemia in adults with an incidence of 2–4/100,000 per year and extremely short survival [4]. Thanks to advances in modern healthcare, the 5-year overall survival of patients with AML has improved, but it accounts for only 30.5% up to 2023 and differs between various age groups, reaching 50% in younger patients and less than 10% in patients older than 60 years [5,6]. The etiology of AML is heterogeneous. In some patients, prior exposure to therapeutic, occupational, or environmental DNA-damaging factors is implicated, but most cases of AML remain without an obvious etiology [4]. Recently, many recurrent somatic mutations in AML have been identified, including early, disease-initiating driver mutations and germline predisposing mutations/variants, as well as co-occurring passenger lesions [7].

AML is diagnosed with a blast threshold ≥ 20% in the bone marrow/blood or >10% in the presence of genetic abnormalities that define specific AML subtypes [8]. The risk stratification of patients diagnosed with AML takes into account numerous disease factors such as the presence/absence of adverse cytogenetic features, germline predisposition, age, poor performance status, prior exposure history to cytotoxic agents or radiotherapy, and prior history of myelodysplasia or myeloproliferative neoplasm, amongst others that are the strongest clinical predictors of early death [9]. Currently, measurable residual disease (MRD) is the most incorporated into longitudinal risk assessments [5,10].

The goal of AML treatment should be the eradication of the disease, whenever possible, accomplished by inducing complete remission (CR) with initial therapy [8]. However, the response to treatment and the overall prognosis in a particular patient are variable, dependent on the several patient- and tumor-specific factors mentioned above. Treatment failure in AML is represented by two concepts: refractory disease and relapsed disease [8,11].

One of the main obstacles in selecting the optimal treatment option for AML therapy is the lack of validated criteria to consider a patient fit or unfit for intensive chemotherapy. The choice of therapy is dictated by many of the same principles that govern risk stratification [12]. Assigning a patient to a particular treatment group based on the presence/absence of target mutations (so-called, risk stratification by genetics) is an important consideration in determining the proper choice of regimen [8,13]. Additionally, managing tactics regarding AML patients may include prognosis stratification by cytogenetic/molecular markers and clinical characteristics [14]. Patients considered fit for intensive therapy are managed with more aggressive induction regimens that include anthracyclines and cytarabine. Patients considered unsuitable for intensive therapy are managed with lower intensity regimens incorporating hypomethylating agents with venetoclax or low-dose cytarabine [15,16].

Despite significant advances in improving the outcomes of hematological disorders over the past decades, mostly due to the development of targeted therapeutics, AML patients respond differently to treatment and prognosis [17]. Recent studies have explored the relevant contributions of clinical, genetic, demographic, and treatment variables to predict event-free and overall survival in patients with AML [8,18]. However, models incorporating all of these factors that aim to predict whether a patient with a definite set of covariates will have a longer remission or life expectancy than a patient with another set of covariates will not always provide the correct results. This emphasizes the necessity of not only evaluating established prognostic factors but also of focusing on new variables and models for the prediction of therapy response and survival in AML patients.

It is known that induction failure due to resistance to chemotherapeutic agents represents a serious obstacle for improving survival outcomes in AML [19,20]. Previously, we have demonstrated that the drug responsiveness of tumor cells in AML patients correlates with the response to therapy as well as with the presence of immunophenotypic and cytogenetic prognostic markers [21]. The prognostic significance of sensitivity or the resistance of leukemic cells to cytotoxic drugs depended on the therapy given to a patient [22].

The present study evaluated the contribution of tumor cell drug responsiveness to predict the therapy response and overall survival of patients with AML. Our study demonstrates that the developed prognostic scale for the risk stratification of AML patients based on cell sensitivity to chemotherapeutic drugs, *MDR1* mRNA expression, tumor origin (primary or secondary), unfavorable cytogenetic abnormalities, and aberrant immunophenotype is related to the therapy response and represents an independent predictive factor for the overall survival of newly diagnosed AML patients without complicated clinical and hematological anamnesis.

## 2. Materials and Methods

### 2.1. Patients

A total of 53 patients with acute myeloid leukemia (AML) admitted to the Novosibirsk Hematology Center from 1 January 2014 to 31 December 2018 were enrolled in this study. The average age of patients was 51.2 ± 14.5 years (Appendix A). The gender distribution of the patients was as follows: male—25 patients (47.2%), female—28 patients (52.8%) (Appendix A). Anamnesis collection, objective examination, laboratory tests (peripheral blood and urine analysis, biochemical tests (total protein, alanine aminotransferase, aspartate aminotransferase, prothrombin time, thymol test, C-reactive protein, fibrinogen, lactate dehydrogenase, alkaline phosphatase, creatinine, urea, uric acid, Na^+^, K^+^, Ca^2+^)), and computed tomography of the abdomen, chest, pelvis, and retroperitoneal space, as well as bone marrow aspiration/biopsy with myelogram counting, multiparameter flow cytometry using a wide panel of primary antibodies to hematopoietic cell differentiation clusters, and cytogenetic analysis with detection of chromosome abnormalities using metaphase plates and the FISH method were performed. All recruited patients were diagnosed according to the 5th Edition of the World Health Organization classification of hematolymphoid tumors [23] and received treatment in accordance with the standard clinical recommendations [8].

Inclusion criteria: patients diagnosed with AML on the base of clinical, morphological, and genetic markers according to the standard diagnostic protocols for AML were included in this study.

Exclusion criteria: patients presenting with hematological disorders (e.g., anemia) because of other non-malignant causes, patients with HIV, hepatitis B and C, and tuberculosis were excluded from this study.

During the study, patients with AML were assigned into two cohorts: patients received intensive induction chemotherapy with anthracycline-based regimens and patients received non-intensive chemotherapy with low-dose cytarabine because of their comorbidity and complicated clinical and hematological anamnesis.

This study was approved by the Institutional Review Board of Novosibirsk State Medical University (protocol No. 80 from 17 December 2015), and informed consent was obtained from the patients. This study was conducted in compliance with the ethical principles of the Declaration of Helsinki in the current version [24].

### 2.2. Collection of Bone Marrow Samples

Bone marrow samples were obtained from AML patients at diagnosis before the start of treatment, as described previously [21,22], and were transported to the laboratory within 3-4 h of material collection.

### 2.3. Biobanking

Bone marrow samples were obtained at the time of diagnosis before the treatment and were stored at −80 °C for further in vitro studies.

### 2.4. Cell Isolation and Culture

Tumor cells were isolated from the bone marrow of AML patients using centrifugation in lymphocyte separation medium (MP Biomedicals, Santa Ana, CA, USA) according to the manufacturer’s protocol and were cultured in the Iscove’s Modified Dulbecco’s Medium (IMDM) supplemented with 10% foetal bovine serum and 1% solution of antibiotics and antimycotic (10,000 μg/mL streptomycin, 10,000 IU/mL penicillin, and 25 μg/mL amphotericin; ICN, Eschwege, Germany) at 37 °C in a 5% CO_2_.

### 2.5. Water-Soluble Tetrazolium (WST)-Test

Tumor cells were seeded in 96-well plates at a density of 10^5^ cells per well and incubated with daunorubicin (0, 0.05, 0.1, 0.2, 0.4, 0.6, 1, and 2 μM) or cytarabine (0, 0.001, 0.01, 0.2, 0.8, 4, 40, and 82 μM) (both from TEVA, Rehovot, Israel) for 72 h at 37 °C. Next, the cells were incubated with WST-1 solution (10 μL in a 0.5 mg/mL, Roche, Basel, Switzerland) for 3 h at 37 °C. The absorbance was measured at 450 and 620 nm with a Multiscan RC spectrophotometer (Labsystems, Vantaa, Finland). The concentration of chemotherapeutic drugs that caused the death of 50% of tumor cells (IC_50_) was calculated as described in [21]. Then, based on IC_50_ values, the drug responsiveness of leukemia cells was scaled for subsequent analysis (Table 1).

### 2.6. Real-Time qPCR

Total RNA was extracted from tumor cells using TRIzol Reagent (Invitrogen, Carlsbad, CA, USA) according to the manufacturer’s protocol. RNA quantification was performed with real-time qPCR using BioMaster HS-qPCR SYBR Blue master mix with SYBR Green I fluorescent dye (Biolabmix, Novosibirsk, Russia) and IQ5 Cycler (Bio-Rad, Hercules, CA, USA). The following primers were used: MDR1_F: 5′-AGAGAATCCCCTCCAGATAAGA-3′, MDR1_R: 5′-AAGCCTATTCCATTTTGAACTTTCT-3′, GAPDH_F: 5′-GTGAAGGTCGGAGTCAAC-3′, and GAPDH_R: 5′-TGGAATTTGCCATGGGTG-3′. All measurements were performed in triplicate. Relative level of gene expression was normalized to the level of GAPDH according to the ∆∆Ct method and was determined with CFX96^TM^ Real-Time system (C1000 Touch^TM^, Hercules, CA, USA). MDR1 mRNA expression levels in tumor cells were assessed as described previously [21].

### 2.7. Immunocytochemistry

Bone marrow smears of AML patients were incubated with anti-P-glycoprotein (P-gp) antibody [EPR10364-57] (ab170904, Abcam, Cambridge, UK) according to the manufacturer’s protocol. UltraVision Quanto Detection System (Thermo Fisher Scientific, Waltham, MA, USA) and Permanent Fast Red Quanto Substrate System (Thermo Fisher Scientific, USA) were used. Finally, samples were counterstained with Romanovsky–Giemsa and visualized using an Axiostar Plus microscope equipped with an Axiocam MRc5 digital camera (Carl Zeiss, Jena, Germany). P-gp staining was assessed as described previously [21].

### 2.8. Statistical Analysis

Correlation analysis was performed using Spearman’s rank order correlation coefficient *r*, which reflects the strength of the statistical relationships between the studied variables. A value of *r* 0.01−0.29 indicates a weak positive correlation, 0.30–0.69—moderate positive correlation, and 0.70−1.0—strong positive correlation.

Overall survival was evaluated using the Kaplan–Meier method, and comparisons between groups were performed using the log-rank test.

Receiver operating characteristic (ROC) analysis was performed to assess the predictive value of our prognostic scale for risk stratification with respect to therapy response and overall survival of AML patients. ROC curves were constructed and the area under the curve (AUC) reflecting model quality was evaluated as follows: 0.9−1.0—excellent, 0.8−0.9—very good, 0.7−0.8—good, 0.6−0.7—moderate, 0.5−0.6—bad. The optimal cutoff was selected using the Youden index, which maximizes the difference between the true positive rate and the false positive rate. The model was considered statistically significant at *p* value < 0.05, area under the curve (AUC) > 0.5, and sensitivity and specificity > 60%.

Univariate and multivariate Cox regression analyses were conducted to estimate the predictive value of our prognostic scale for risk stratification and other studied variables with respect to the overall survival of AML patients.

Statistical analysis was performed using MS Excel, OriginPro 7.5, Statistica v13.1, and IBM SPSS software platform. All results with *p* value ≤ 0.05 were considered statistically significant.

## 3. Results

### 3.1. Characteristics of AML Patients

#### 3.1.1. Comorbidity

This study included 53 patients with acute myeloid leukemia (AML). To assess the general status of the patients and the aggressiveness of the tumor process, anamnesis, objective examination, and laboratory tests were performed. Before the treatment was started, all patients underwent computed tomography and bone marrow aspiration/biopsy with myelogram counting. Moreover, multiparameter flow cytometry and cytogenetic analysis with an evaluation of leukemia-associated immunophenotype and chromosome abnormalities (mutations and rearrangements) of tumor cells were carried out to define the disease and risk categories as well as to plan treatment modalities.

The clinical and disease characteristics of the AML patients enrolled in this study are displayed in Appendix A. On the objective examination, lymphadenopathy was revealed in 17% of the AML patients. Most patients did not have enlarged lymph nodes (Appendix A). In the computed tomography, 67.9% of the patients had extranodal lesions, 56.6% of them were represented by specific liver injury, 9.4% by specific lung injury, and 1.9% by skin injury (Appendix A).

At the onset of the disease before the start of treatment, the following clinical syndromes were identified in the studied cohorts of the AML patients: hemorrhagic—79.2%, anemic—94.3%, infectious—50.9%, hyperplastic—66%, and intoxication—86.8% (Appendix A). These syndromes were combined differently in the patients.

Assessing the initial parameters of peripheral blood in AML patients, the following changes were revealed. A normal hemoglobin level (143.3 ± 27 g/L) was observed in only 5.6% of the patients (Appendix A). Anemia was observed in 50 patients (94.3%): mild anemia (Hb—101.2 ± 7.3 g/L) in 34%, moderate anemia (Hb—80.1 ± 6.5 g/L) in 34%, and severe anemia (Hb—55.9 ± 8.5 g/L) in 26.3% of the patients. All anemias were normochromic. Normal platelet counts (213.6 ± 59.4 × 10^9^/L) were found in 13.2% of AML patients. Thrombocytosis (549 ± 9.9 × 10^9^/L) was detected in 3.8% and decreased platelet counts (35.1 ± 28.8 × 10^9^/L) in 83% of the patients. The normal level of leukocytes was detected in 13.2%, leukopenia with mean values of 2 ± 0.5 × 10^9^/L in 15.1%, leukocytosis (44.3 ± 26.2 × 10^9^/L) in 49.1%, and hyperleukocytosis (182.4 ± 119.5 × 10^9^/L) in 22.6% of the patients. Blasts in peripheral blood were found in 79.2% of AML patients, with the average content of blasts amounting to 55.7 ± 29.3%.

#### 3.1.2. Risk and Prognosis Stratification of AML Patients

In 43 out of 53 patients with AML, the karyotype and genetic mutations were assessed. In 10 patients, the results of the cytogenetic study were not obtained, since the number of metaphases was not enough to conduct the analysis. According to the presence or absence of established cytogenetic abnormalities, risk stratification by genetics, according to 2022 European LeukemiaNet (ELN) recommendations for the diagnosis and management of AML in adults [8], was determined for each studied patient. In this way, patients were assigned to risk categories: favorable, intermediate, and adverse (Appendix A). Most of the patients enrolled in this study (53.5%) were assigned to the favorable risk category. The gender distribution among the favorable risk group was as follows: male—34.9%, female—18.6%. The intermediate risk group included 11.6% of the AML patients: male—4.6%, female—7%. The adverse risk group included 34.9% of the patients: male—11.6%, female—23.3%. Thus, male patients predominated in the favorable risk group and female patients predominated in the intermediate and unfavorable risk groups.

In all the AML patients (*n* = 53), according to the combination of established clinical and molecular markers (age, the level of leukocytes, tumor status (primary or secondary leukemia), cell immunophenotype, extramedullary lesions, measurable residual disease (MRD), and particular cytogenetic mutations), prognosis stratification using an integrated-risk adapted approach [14] with modifications was performed, and a prognosis category (favorable or unfavorable) for each patient was determined (Appendix A). Thus, in attributing to the prognosis group based on the cytogenetic/molecular markers with an additional assessment of clinical characteristics, most of the patients (81.1%) had a poor prognosis with an approximately equal gender distribution.

#### 3.1.3. Therapy Response

Responses to antitumor therapy were assessed in each patient with AML after 1–2 courses of induction chemotherapy according to [8] and were as follows: complete remission—bone marrow blasts < 5%, absence of circulating blasts, absence of extramedullary disease, granulocyte count ≥ 1 × 10^9^/L, platelet count ≥ 100 × 10^9^/L; relapsed disease—bone marrow blasts ≥ 5%, reappearance of blasts in peripheral blood, or development of extramedullary disease; primary chemoresistance (refractory disease)—no response after 1–2 courses of intensive induction treatment.

As can be seen from the data presented in Appendix A, most patients demonstrated primary chemoresistance (60.4%). Among these patients, males and females were presented approximately equally. Remission and relapse were achieved in 30.2% and 9.4% of all the AML patients, respectively. In these groups of patients, females predominated. Thus, it can be assumed that some patients classified as a favorable risk category were initially resistant to induction chemotherapy (compare data in Appendix A).

### 3.2. Prognostic Factors Affecting the Therapy Response in AML Patients; Correlation Analysis

Previously, we studied the relationships between the drug responsiveness of tumor cells as well as immunologic and cytogenetic markers and the therapy response and overall survival of leukemia patients [21,22]. Our data clearly demonstrate that the therapy response of AML patients along with the established prognostic markers correlates well with the sensitivity of their tumor cells to cytotoxic drugs detected in vitro. However, in accordance with the current recommendations for the diagnosis and management of AML patients, important prognostic factors together with the immunophenotype and cytogenetic abnormalities, are age, leukocyte count, MRD, and the presence of neuroleukemia [8]. In the present study, we compared the impact of established prognostic factors and parameters reflecting the multidrug resistance (MDR) phenotype of tumor cells (drug responsiveness, *MDR1* mRNA/P-gp expression) on the therapy response and, as a result, the prognosis for AML patients. In addition to the standard prognostic factors mentioned above, clinical syndromes (hemorrhagic, infectious, hyperplastic, anemic, intoxication) and the origin of leukemia (primary/secondary) were added to the analysis. Correlations between the therapy response and all the studied variables were assessed and are presented in Table 2 and Table 3. Patients with AML who received induction therapy with intensive anthracycline-based regimens (cohort 1) and those who received non-intensive therapy with low-dose cytarabine (cohort 2) were analyzed separately.

#### 3.2.1. AML Patients Received Intensive Therapy with Anthracycline-Based Regimens (Cohort 1)

In cohort 1 of the AML patients, statistically significant (*p* ≤ 0.05) positive correlations of the therapy response were identified with the following parameters: strong correlations—with cell sensitivity to daunorubicin (*r* = 0.72); moderate correlations—with total cell sensitivity to cytotoxic drugs (*r* = 0.61), age (*r* = 0.55), tumor cell karyotype (*r* = 0.60), risk stratification by genetics (*r* = 0.70), prognosis stratification by cytogenetic/molecular markers, and clinical characteristics (hereinafter, standard prognosis stratification) (*r* = 0.55) (Table 2). A strong positive correlation between the therapy response and the expression levels of *MDR1* mRNA (*r* = 1.0) was also found; however, the *p* value was >0.05.

As for drug responsiveness, statistically significant positive correlations of the sensitivity of tumor cells to daunorubicin were identified with the following variables: strong—with therapy response (*r* = 0.72), total sensitivity to cytotoxic drugs (*r* = 0.87); moderate—with sensitivity to cytarabine (*r* = 0.56), cytogenetic abnormalities (*r* = 0.54), risk stratification by genetics (*r* = 0.63), standard prognosis stratification (*r* = 0.52) (Table 2). Significant positive correlations for the sensitivity of tumor cells to cytarabine were identified: strong—with total sensitivity to cytotoxic drugs (*r* = 0.83); moderate—with sensitivity to daunorubicin (*r* = 0.56) (Table 2).

Moreover, strong positive correlations were found between *MDR1* mRNA expression levels in the tumor cells of AML patients in cohort 1 and response to therapy (*r* = 1.0), as well as aberrant immunophenotype of tumor cells (*r* = 0.73) and standard prognosis stratification (*r* = 1.0); however, *p* was >0.05. A significant moderate positive correlation was also found between P-gp expression and the origin of the disease (primary/secondary AML) (*r* = 0.59) (Table 2).

#### 3.2.2. AML Patients Received Non-Intensive Therapy with Low-Dose Cytarabine (Cohort 2)

In cohort 2 of AML patients, statistically significant (*p* ≤ 0.05) strong and moderate positive correlations between the therapy response, the sensitivity of tumor cells to cytarabine (*r* = 0.80), and the total sensitivity to cytotoxic drugs (*r* = 0.57) were found (Table 3).

Significant positive correlations of tumor cell sensitivity to daunorubicin were identified with sensitivity to cytarabine (*r* = 0.60) and total sensitivity to chemotherapeutic drugs (*r* = 0.96). Significant positive correlations of cell sensitivity to cytarabine were found with therapy response (*r* = 0.80), sensitivity to daunorubicin (*r* = 0.60), total sensitivity to cytotoxic drugs (*r* = 0.79), and the presence of hyperplastic syndrome (*r* = 0.62). There were no statistically significant correlations between *MDR1* mRNA and P-gp expression levels and the studied parameters (Table 3).

Thus, among the analyzed prognostic factors affecting the therapy response of AML patients, the sensitivity of tumor cells to chemotherapeutic drugs in vitro and the karyotype of tumor cells may have the greatest prognostic value.

### 3.3. Effective Prognostic Model for Therapy Response Prediction in AML Patients

Currently, the generally accepted practice is to evaluate a set of patient- and disease-associated factors, according to which the patient is assigned to a risk group predicting therapy response and disease outcome [8,14]. However, the list of these parameters does not include the factors reflecting the sensitivity of patients’ tumor cells to cytotoxic drugs as well as other parameters of the MDR phenotype that significantly affect patient response to antitumor therapy [25].

Based on the presented data on the close interactions between the drug responsiveness of tumor cells and the response to antitumor therapy, we developed a prognostic scale based on the evaluation of the sensitivity of tumor cells to chemotherapeutic drugs and the expression of *MDR1* mRNA in tumor cells, as well as standard prognostic factors (origin of leukemia, karyotype, age, presence of aberrant immunophenotype) for the risk stratification of AML patients (Table 4). Previously, all patients were divided into the groups according to (1) the sensitivity of their tumor cells to cytotoxic drugs (high, moderate, low), (2) the level of *MDR1* mRNA expression (weak, moderate, strong), (3) the presence of one or more unfavorable mutations in the tumor cells, (4) the age of the patient (<40 years, 40−60 years, >60 years), and (5) the presence of one or more markers of aberrant immunophenotype in the tumor cells. As a result of the sum of the scores in all the parameters of our prognostic scale, patients can be divided into three risk groups: 0−2—low risk group, 3−5—intermediate risk group, 6−12—high risk group, which makes it possible to predict the response of a particular patient to the selected chemotherapy protocol before the treatment start.

Next, we compared our prognostic scale, risk stratification by genetics, and standard prognosis stratification according to their correlations with the response to therapy in AML patients. As performed previously, for this analysis, all patients were divided into the same cohorts: patients received intensive induction therapy with anthracyclines (cohort 1) and patients received non-intensive therapy with low-dose cytarabine (cohort 2). Data comparing the effectiveness of our prognostic scale with the existing ones are presented in Table 5.

In cohort 1, the distribution of AML patients by risk groups according to our prognostic scale correlated best with the response to therapy (*r* = 0.84). The distribution of patients according to the karyotype of their tumor cells (risk stratification by genetics) is in second place in terms of correlation with the therapy response (*r* = 0.67). The distribution of patients according to the currently existing prognostic groups (prognosis stratification by cytogenetic/molecular markers and clinical characteristics) is in the third position with *r* = 0.56 (Table 5). In cohort 2, only our prognostic scale was correlated as statistically significant with the response to therapy (*r* = 0.54). Correlations between risk stratification by genetics, standard prognosis stratification, and therapy response were not detected in this cohort (Table 5). Thus, the risk stratification of AML patients according to the developed prognostic scale significantly correlates with their therapy response and can be predictively effective.

### 3.4. Predictive Value of Our Prognostic Model in AML Patients

#### 3.4.1. Survival and ROC Analysis

The survival rates of AML patients assigned to the risk groups according to our prognostic scale were evaluated (Figure 1). The overall survival of patients in the low-risk group was 58% (6-month and 1-year survival) and 46% (3-year survival) with a medium survival rate of 452 days. The overall survival of patients classified as the intermediate risk group was as follows: 6-month survival 53%, 1-year survival 28%, and 3-year survival 12% with a medium survival rate of 239 days. Patients assigned to the high-risk group demonstrated 6-month survival 30%, 1-year survival 25%, and 3-year survival 10%, with a medium survival rate of 79 days. Thus, the survival of patients assigned to the low-risk group was quite higher than that of patients assigned to the high- and intermediate-risk groups according to our prognostic scale, which indicates the reliability and robustness of our classification.

To evaluate the predictive value of our prognostic model, receiver operating characteristic (ROC) analysis was performed. ROC curves were constructed to assess the sensitivity and specificity of our prognostic scale with respect to therapy response and the overall survival of AML patients, and area under the curve (AUC) was used to evaluate the prognostic efficacy of our model. For ROC curves, AUC > 0.9 suggests that the accuracy of the model is very high; an AUC value between 0.7 and 0.9 indicates that the model has a certain accuracy (0.8−0.9—very good, 0.7−0.8—good); an AUC value between 0.5 and 0.7 shows that the accuracy of the model is general [26]. The therapy response was defined as described previously (see section). Overall survival was defined as the time from study enrollment to death, with living patients censored on the date of the last follow-up [27].

According to the AUC values of the ROC curves, the accuracy of the developed prognostic model is very good in regard to therapy response and good in regard to overall survival of AML patients. Figure 2 demonstrates that the probability of achieving or not achieving the response to therapy in patients with AML was highly associated with the risk group based on our prognostic scale: the AUC was 0.816 ± 0.069 (95% CI 0.681–0.950, *p* < 0.0001) with a sensitivity of 97.4% and specificity of 60%. The overall survival of patients with AML was also closely related to our prognostic scale, with an AUC of 0.773 ± 0.088 (95% CI 0.600–0.947, sensitivity—97.7%, specificity—80%, *p* < 0.01). Thus, it can be assumed that our prognostic scale for risk stratification possesses a certain predictive value for AML patients.

#### 3.4.2. COX Regression Analysis

The wide variety of factors associated with the outcome of the disease makes it necessary to use the Cox proportional hazards regression model, which allow us to include in the analysis a large set of variables affecting or presumably affecting the outcome [28]. Therefore, at the final step of our study, the influence of risk stratification according to our prognostic scale as well as the established prognostic factors (variables) on the overall survival of AML patients using univariate and multivariate Cox proportional hazards regression models was analyzed.

The univariate Cox proportional hazards model enables us to assess the possible impact of risk stratification according to our prognostic scale on AML patients’ survival as well as evaluate the influence of each prognostic factor separately and compare them with each other. The multivariate Cox proportional hazards model allows us to assess the simultaneous influence of more than one factor on the outcome (overall survival) of AML patients, find the interactions of these variables among themselves, and evaluate the independent prognostic effect of the study covariates, in particular the risk stratification of the patients according to our prognostic scale. The output parameters of the Cox regression models are the hazard ratio (HR) and its confidence interval (95% CI). The HR value (greater than 1 with statistically significant results) in this case is equal to the chance that a patient of a high-risk cohort will achieve an outcome earlier than a patient of a lower-risk cohort. P-values are used in Cox regression models to exclude redundant or unnecessary variables. The patients of cohorts 1 and 2 were analyzed separately.

In cohort 1, in the univariate analysis, the risk group defined in accordance with the risk stratification by genetics (HR 1.96, 95% CI 1.03–3.68, *p* = 0.039) and the risk group determined by our prognostic scale (HR 1.17, 95% CI 1.17–3.93, *p* = 0.01) demonstrated an influence on the overall survival of leukemia patients (Table 6). In the multivariate analysis of this model (*p* value of the model = 0.035), the impact on the overall survival of AML patients accounting for the influence of other prognostic variables presented in Table 6 was exerted by the risk group determined by our prognostic scale (HR 2.8, 95% CI 1.31–6.02, *p* = 0.008).

In cohort 2, following univariate analysis, only the origin of the tumor (primary or secondary AML) demonstrated an influence on the overall survival of leukemia patients, however, with low values of HR (HR 0.34, 95% CI 0.14–0.79, *p* = 0.01), which indicates a relatively small impact of this variable on the disease outcome (Table 7). In the multivariate analysis of this model (p value of the model = 0.01), the influence on the overall survival of AML patients accounting for the influence of other prognostic variables presented in Table 7 was exerted by the tumor origin (HR 0.37, 95% CI 0.13–0.99, *p* = 0.049), the sensitivity of tumor cells to cytarabine (HR 5.8, 95% CI 1.29–26.05, *p* = 0.02), and the level of leukocytes in peripheral blood (HR 6.38, 95% CI 1.64–26.1, *p* = 0.008). It should be noted that the HR values for tumor origin were also low as in the univariate analysis.

Thus, univariate and multivariate Cox proportional hazards regression analyses clearly demonstrate that the risk stratification according to the prognostic scale introduced in this work is an independent predictive factor for the overall survival of primary AML patients without complicated clinical and hematological anamnesis, severe comorbidities, and previous treatment.

## 4. Discussion

Progress in acute myeloid leukemia (AML) treatment is occurring at an unprecedented pace. The past decades have been marked by an increasing progress in the scientific understanding of AML biology, leading to enhanced prognostication tools and risk assessments [5,29]. Due to a combination of advances in supportive care and the availability of novel, often molecularly targeted, therapies, the life expectancy of AML patients continues to improve [30,31]. However, despite obvious progress in the diagnosis and management of AML, some obstacles in the successful achievement of the therapy response exist, one of which is the initial resistance of leukemic cells to cytotoxic drugs [3,32]. Therefore, an initial assessment of the sensitivity of tumor cells to chemotherapeutic agents used in the induction therapy of leukemia and other parameters reflecting the multidrug resistance phenotype, such as drug efflux transporters, may be useful in predicting the patient’s response to antitumor therapy. Previously, in our studies related to acute leukemias [21,22], as well as in the works related to a wide range of neoplastic diseases of various histological origins (gastric cancer [33], renal cancer [34], colorectal cancer [35], osteosarcoma [36]), correlations of the therapy response and the clinical outcome of tumor patients with the drug sensitivity of their tumor cells were demonstrated. As for hematological disorders and particularly AML, many works and reviews show the clinical and prognostic relevance of drug transporter proteins [37,38,39,40,41]. However, in most studies, drug resistance marker characteristics were considered separately.

Currently, the main approach for predicting the therapy response and overall survival of AML patients is represented in the risk stratification by genetics based on the cytogenetic abnormalities in leukemic cells [8]. However, it should be emphasized that, for AML patients fit for allogeneic hematopoietic cell transplantation (HCT), decision-making information takes into account the complexity of different risk factors: disease-specific risk factors (e.g., chromosomal aberrations and gene expression profiles), transplantation-specific risk factors (e.g., the choice of donor grafts or stem-cell source), and in some cases patient-specific risk factors (e.g., age and compromised organ functions (comorbidities)), which guide the choice of appropriate HCT regimen [42]. In this way, a distribution of patients into risk categories (favorable, intermediate, adverse) according to the risk stratification by genetics is complemented by the clinical scoring systems, such as the Hematopoietic Cell Transplantation Comorbidity Index (HCT-CI) and the Disease Risk Index (DRI), which focus on particular contributing risk factors, such as comorbid conditions or the course of the underlying disease, as well as European Group for Blood and Marrow Transplantation (EBMT), Acute Leukemia (AL)-EMBT, and Pretransplantation Assessment of Mortality (PAM) scores, which take a more holistic approach, seeking to explain the variability of outcomes in broader terms [43,44].

Recently, similar combined scoring systems for survival and prognosis prediction have also appeared for patients with AML who are not eligible for HCT. Silveira et al. demonstrated that the distribution of AML patients in accordance with the created novel scoring system integrating both cytogenetic/molecular information and clinical prognostic features such as age (>45 years), white blood cell count (<1.5 or >30.0 × 10^3^/μL), and low albumin levels (<3.8 g/dL) were associated with worse overall survival in test cohorts [45]. In the study of Tsai et al., the incorporation of long non-coding RNA (lncRNA) expression profiles in the 2017 ELN risk classification by genetics was shown to improve the prognostic prediction of AML patients: higher lncRNA scores were significantly associated with older age, adverse gene mutations, and shorter overall and disease-free survival [46]. Such a combined scoring approach can be useful for informed decision making, better patient counseling, the design of interventional trials, treatment allocation, and personalization according to predicted risk for AML patients. 

In our study, we attempted to enhance the survival and therapy response prediction for AML patients through a combined approach that incorporates information on drug responsiveness and the *MDR1* mRNA/P-gp expression of leukemic cells at diagnosis into the standard protocol for risk and prognosis stratification. Primary chemoresistance to the induction therapy of several AML patients assigned to the favorable risk category (Appendix A) may indicate an insufficient predictive value of the risk stratification by genetics, whereas the distribution of the patients into risk groups, considering the indicators of MDR phenotype such as sensitivity to chemotherapeutic drugs and *MDR1* mRNA/P-gp expression in their tumor cells, demonstrated a stronger correlation with response to therapy and the survival of leukemia patients (Table 5, Table 6, Table 7, Figure 2). It is noteworthy that the consideration of the sensitivity of leukemic cells to only two cytotoxic drugs used in both intensive and non-intensive induction therapy schemes (daunorubicin and cytarabine) improved the predictive power of standard scoring systems for the ranging of AML patients.

In the last few years, a lot of information, based mainly on the computational approach concerning the predictive role of individual genetic and molecular markers for leukemia patients resistant to chemotherapy, has appeared. For example, in the study of Xu et al., tight associations between the high expression of TSC22D3 domain family genes playing an essential role in tumor progression with poor overall and event-free survival and drug resistance to BCL2 inhibitors in adult AML patients, especially in the chemotherapy group, have been reported [47]. The study of Liu et al. demonstrated that high miR-107 or miR-17 expression levels were associated with poorer overall survival and event-free survival in the chemotherapy cohort of patients with de novo AML [48]. A special place is occupied by immunity-associated biomarkers, such as immune-related long noncoding RNAs [49] and immune microenvironment genes [50], which are closely related to resistance to immunotherapy and poor prognosis for AML patients. Such molecular and genetic features are designed to complement the risk stratification by genetics and represent high-tech approaches with high cost and the need for specially trained staff. The evaluation of the drug responsiveness of the leukemic cells in vitro does not require expensive equipment, takes a short time, and in general can be useful for a preliminary assessment of the response of AML patients to a number of components of standard induction chemotherapy (both intensive and non-intensive), which may be of particular value and applicability in low- and middle-income countries that are limited in full cytogenetic and molecular information at diagnosis and can be one of the steps towards personalized therapy for leukemia patients.

## 5. Conclusions

Obvious progress in the diagnosis and management of AML in recent years has been observed. However, a number of obstacles in the successful achievement of the therapy response of AML patients continue to exist, one of which is the initial resistance of leukemic cells to cytotoxic drugs. Our study demonstrates that the designed prognostic scale for the risk stratification of AML patients based on the sensitivity of tumor cells to chemotherapeutic drugs is closely related to the therapy response and overall survival of newly diagnosed AML patients without complicated clinical and hematological anamnesis. Moreover, the stratification of patients with AML into risk groups according to our scale was shown to be an independent predictor for patient survival, and the initial evaluation of the drug responsiveness of tumor cells at diagnosis may be a feasible approach to assign patients to the most effective first-line treatment options.

## 6. Limitations

The limited number of patients enrolled in this study may introduce a certain bias into our results and the need to add more samples for better statistical power of the research.

## Figures and Tables

**Figure 1 jpm-13-01234-f001:**
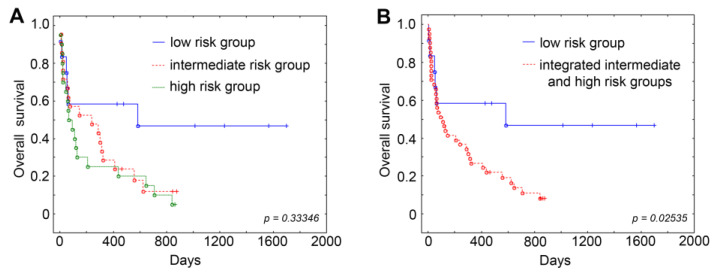
The Kaplan–Meier survival curves for AML patients assigned to the low, intermediate and high risk groups (**A**) and for AML patients assigned to the low and integrated intermediate and high risk groups (**B**) according to our prognostic scale.

**Figure 2 jpm-13-01234-f002:**
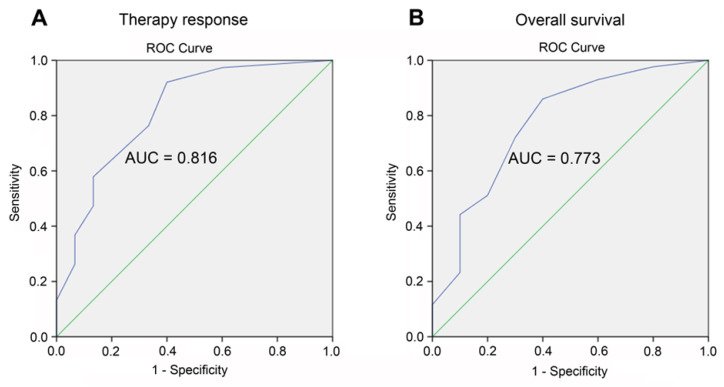
Receiver operating characteristic (ROC)-curves demonstrated the predictive value of our prognostic scale for risk stratification in relation to the therapy response ((**A**), sensitivity 97.4%, specificity 60%, *p* < 0.0001) and overall survival ((**B**), sensitivity 97.7%, specificity 80%, *p* < 0.01) of AML patients. The model was considered statistically significant at *p* value < 0.05, area under the curve (AUC) > 0.5, and sensitivity and specificity > 60%.

**Table 1 jpm-13-01234-t001:** Scales for the distribution of AML patients according to the sensitivity of their tumor cells to chemotherapeutic drugs.

Chemotherapeutic Drug	IC_50_ Values		
	1—High Sensitivity	2—Moderate Sensitivity	3—Low Sensitivity
Daunorubicin, μM	0−0.25	0.25−0.5	>0.5
Cytarabine, μM	0−1.5	1.5−8	>8

**Table 2 jpm-13-01234-t002:** Correlations between factors affecting the therapy response in AML patients who received intensive therapy with anthracyclin-based regimens.

	Therapy Response	Sensitivity to Daunorubicin	Sensitivity to Cytarabine	*MDR1* mRNA Expression	P-gp Expression
Therapy response		0.72 *	0.43	1.0	0.36
Sensitivity to daunorubicin	0.72 *		0.56 *	0.58	0.52
Sensitivity to cytarabine	0.43	0.56 *		0.33	−0.26
Total sensitivity to chemotherapeutic drugs	0.61 *	0.87 *	0.83 *	0.61	0.18
*MDR1* mRNA expression	1.0	0.58	0.33		0.55
P-gp expression	0.36	0.52	−0.26	0.55	
Aberrant immunophenotype	0.31	0.39	0.24	0.73	0.30
Age	0.55 *	0.39	0.25	0.65	−0.15
Cytogenetic abnormalities	0.60 *	0.54 *	0.38	0.15	0.05
Primary/Secondary AML	0.34	0.29	−0.07	0.33	0.59 *
Risk stratification by genetics	0.70 *	0.63 *	0.30	0.58	0.17
Prognosis stratification by cytogenetic/molecular markers and clinical characteristics	0.55 *	0.52 *	0.25	1.0	0.24
CD34	0.02	−0.41	−0.22	0.45	0.10
HLA-DR	0.39	0.26	0.48	0.65	−0.12
Leukocytes	0.09	0.13	0.19	0.58	−0.36
Hemoglobin	−0.10	−0.41	−0.17	0.44	−0.02
Platelets	−0.30	−0.49 *	−0.23	−0.45	−0.42
Lactate dehydrogenase	−0.08	0.36	0.39	0.45	−0.10
Hemorrhagic syndrome	−0.42 *	−0.27	−0.04	−0.33	−0.29
Infectious syndrome	0.01	0.01	−0.07	0.33	−0.15
Intoxication	0.08	−0.09	−0.25	0.65	0.04
Hyperplastic syndrome	−0.05	−0.07	0.08	0.45	−0.05
Neuroleukemia	−0.18	−0.18	0.18	−0.05	−0.20

Red color denotes strong positive correlations (0.70 ≤ *r* ≤ 1.00), blue color denotes moderate positive correlations (0.30 ≤ *r* ≤ 0.69), green color denotes weak positive correlations (0.01 ≤ *r* ≤ 0.29), and grey color denotes uninformative correlations. * indicates statistically significant differences at *p* ≤ 0.05.

**Table 3 jpm-13-01234-t003:** Correlations between factors affecting the therapy response in AML patients who received non-intensive therapy with low-dose cytarabine.

	Therapy Response	Sensitivity to Daunorubicin	Sensitivity to Cytarabine	*MDR1* mRNA Expression	P-gp Expression
Therapy response		0.38	0.80 *	-	0.04
Sensitivity to daunorubicin	0.38		0.60 *	0.47	0.17
Sensitivity to cytarabine	0.80 *	0.60 *		-	-
Total sensitivity to chemotherapeutic drugs	0.57 *	0.96 *	0.79 *	0.49	0.07
*MDR1* mRNA expression	-	0.47	-		-
P-gp expression	0.04	0.17	-	-	
Aberrant immunophenotype	0.18	0.22	0.24	0.31	0.04
Age	0.33	0.27	0.21	−0.10	0.32
Cytogenetic abnormalities	0.22	0.05	0.45	−0.20	−0.17
Primary/Secondary AML	0.21	−0.04	0.38	−0.22	0.09
Risk stratification by genetics	0.09	−0.14	0.35	−0.42	−0.08
CD34	0.29	−0.04	0.19	-	0.27
HLA-DR	−0.33	0.22	−0.33	0.61	−0.50
Leukocytes	−0.22	−0.63 *	−0.42	0.09	−0.15
Hemoglobin	0.31	−0.04	−0.11	0.27	−0.08
Platelets	0.02	−0.20	−0.19	0.09	0.09
Lactate dehydrogenase	−0.13	0.23	−0.12	0.29	0.10
Hemorrhagic syndrome	−0.29	0.06	−0.25	0.65	−0.61
Infectious syndrome	0.05	0.09	−0.02	0.65	−0.36
Intoxication	0.45	−0.24	−0.13	-	-
Hyperplastic syndrome	0.21	0.32	0.62 *	-	−0.41

Red color denotes strong positive correlations (0.70 ≤ *r* ≤ 1.00), blue color denotes moderate positive correlations (0.30 ≤ *r* ≤ 0.69), green color denotes weak positive correlations (0.01 ≤ *r* ≤ 0.29), and grey color denotes uninformative correlations. * indicates statistically significant differences at *p* ≤ 0.05. - Data not available.

**Table 4 jpm-13-01234-t004:** Our prognostic scale for risk stratification in AML patients.

Parameter	Formula	Score		
Sensitivity to chemotherapeutic drugs	* High sensitivity—0, moderate sensitivity—1, low sensitivity—2.	0	1	2
*MDR1* mRNA expression	Weak expression—0, moderate expression—1, strong expression—2.	0	1	2
Primary/Secondary AML	Primary—0, secondary—2.	0	-	2
Cytogenetic abnormalities	No mutations—0, one unfavorable mutation—1, more than one unfavorable mutation—2.	0	1	2
Age	<40 years—0, 40–60 years—1, >60 years—2.	0	1	2
Aberrant immunophenotype	The absence of aberrant immunophenotype markers—0, the presence of one of the aberrant immunophenotype markers—1, more than one of the aberrant immunophenotype markers—2.	0	1	2
Total		0	5	12

* The sum of the scores of the sensitivity of tumor cells to each drug is divided by their number.

**Table 5 jpm-13-01234-t005:** Correlations between our risk stratification, risk stratification by genetics, prognosis stratification by cytogenetic/molecular markers/clinical characteristics, and therapy response in AML patients.

Parameter	Therapy Response	Risk Stratification by Genetics	Prognosis Stratification by Cytogenetic/Molecular Markers and Clinical Characteristics	Our Risk Stratification
*Intensive therapy with anthracyclin-based regimens*				
Therapy response		0.67 *	0.56 *	0.84 *
Risk stratification by genetics	0.67 *		0.46 *	0.65 *
Prognosis stratification by cytogenetic/molecular markers and clinical characteristics	0.56 *	0.46 *		0.56 *
Our risk stratification	0.84 *	0.65 *	0.56 *	
*Non-intensive therapy with low-dose cytarabine*				
Therapy response		0.09	-	0.54 *
Risk stratification by genetics	0.09		-	0.20
Our risk stratification	0.54 *	0.20	-	

Red color denotes strong positive correlations (0.70 ≤ *r* ≤ 1.00), blue color denotes moderate positive correlations (0.30 ≤ *r* ≤ 0.69), green color denotes weak positive correlations (0.01 ≤ *r* ≤ 0.29), and grey color denotes uninformative correlations. * indicates statistically significant differences at *p* ≤ 0.05. - Data not available.

**Table 6 jpm-13-01234-t006:** Univariate and multivariate Cox regression analyses of our prognostic scale for risk stratification and other variables in relation to the overall survival of AML patients who received anthracycline-based regimens.

	Univariate Analysis		Multivariate Analysis (*p* = 0.035)	
	*p* Value	HR * (95% CI)	*p* Value	HR * (95% CI)
Age	0.09	1.7 (0.92−3.14)	-	-
Sensitivity to daunorubicin	0.42	1.23 (0.69−2.39)	-	-
Sensitivity to cytarabine	0.39	0.74 (0.38−1.46)	-	-
*MDR1* mRNA expression	0.35	1.76 (0.53−5.8)	-	-
P-glycoprotein expression	0.1	1.03 (0.58−1.83)	-	-
Aberrant immunophenotype	0.7	0.9 (0.42−1.8)	-	-
Cytogenetic abnormalities	0.54	1.17 (0.7−2)	-	-
Leukocytes	0.75	0.9 (0.47−1.71)	0.48	1.36 (0.58−3.23)
CD34	0.38	0.77 (0.43−1.39)	-	-
HLA-DR	0.22	1.87 (0.68−5.17)	-	-
Primary/Secondary AML	0.3	1.29 (0.8−2.07)	-	-
Risk stratification by genetics	0.039	1.96 (1.03−3.68)	-	-
Prognosis stratification by cytogenetic/molecular markers and clinical characteristics	0.12	0.8 (0.77−9.39)	-	-
Our risk stratification	0.01	1.17 (1.17−3.93)	0.008	2.8 (1.31−6.02)
Hemoglobin	0.84	1.06 (0.6−1.88)	0.77	1.1 (0.56−2.19)
Platelets	0.17	0.68 (0.39−1.18)	-	-
Lactate dehydrogenase	0.22	0.55 (0.21−1.44)	-	-
Hemorrhagic syndrome	0.56	0.84 (0.45−1.55)	-	-
Infectious syndrome	0.5	0.73 (0.73−1.88)	0.69	1.1 (0.66−1.9)
Intoxication	0.11	2.28 (0.82−6.28)	0.14	2.27 (0.77−6.65)
Hyperplastic syndrome	0.37	0.81 (0.52−1.29)	-	-
Total sensitivity to chemotherapeutic drugs	0.8	1.09 (0.5−2.36)	-	-

* Hazard ratios (HR) for overall survival according to the univariate and multivariate Cox regression models. - Data not available. Red color indicates statistically significant differences at *p* ≤ 0.05.

**Table 7 jpm-13-01234-t007:** Univariate and multivariate Cox regression analyses of our prognostic scale for risk stratification and other variables in relation to the overall survival of AML patients who received low-dose cytarabine.

	Univariate Analysis		Multivariate Analysis (*p* = 0.01)	
	*p* Value	HR * (95% CI)	*p* Value	HR * (95% CI)
Age	0.98	0.99 (0.43−2.25)	-	-
Sensitivity to daunorubicin	0.3	0.74 (0.41−1.32)	-	-
Sensitivity to cytarabine	0.99	1 (0.32−3.1)	0.02	5.8 (1.29−26.05)
*MDR1* mRNA expression	0.92	1 (0.29−3.99)	-	-
P-glycoprotein expression	0.83	0.91 (0.38−2.17)	-	-
Aberrant immunophenotype	0.83	0.91 (0.38−2.17)	-	-
Cytogenetic abnormalities	0.42	1.3 (0.68−2.53)	-	-
Leukocytes	0.08	1.8 (0.93−3.5)	0.008	6.8 (1.64−26.1)
CD34	0.12	0.46 (0.17−1.23)	-	-
HLA-DR	0.99	1 (0.48−2.1)	-	-
Primary/Secondary AML	0.01	0.34 (0.14−0.79)	0.049	0.37 (0.13−0.99)
Risk stratification by genetics	0.24	1.53 (0.75−3.1)	-	-
Prognosis stratification by cytogenetic/molecular markers and clinical characteristics	-	–	-	-
Our risk stratification	0.72	0.87 (0.4−1.83)	-	-
Hemoglobin	0.66	1.15 (0.62−2.11)	0.33	1.52 (0.65−3.57)
Platelets	0.38	1.4 (0.67−2.9)	0.057	3.5 (0.96−12.73)
Lactate dehydrogenase	0.92	1 (0.35−3.2)	-	-
Hemorrhagic syndrome	0.53	0.65 (0.17−2.5)	-	-
Infectious syndrome	0.095	2.68 (0.84−8.5)	0.67	0.7 (0.14−3.57)
Intoxication	-	–	-	-
Hyperplastic syndrome	0.83	0.85 (0.19−3.86)	-	-
Total sensitivity to chemotherapeutic drugs	0.53	0.78 (0.36−1.7)	-	-

* Hazard ratios (HR) for overall survival according to the univariate and multivariate Cox regression models. - Data not available. Red color indicates statistically significant differences at *p* ≤ 0.05.

## Data Availability

Not applicable.

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
