# Peer review of "Effective Prognostic Model for Therapy Response Prediction in Acute Myeloid Leukemia Patients"

_jpm, 2023, doi:10.3390/jpm13081234_

Round 1

Reviewer 1 Report

The research article “Effective prognostic model for therapy response prediction in acute myeloid leukemia patients” is an attempt to find better first line therapy for effective treatment. They proposed a new prognostic scale for risk stratification of patients with AML based on the detection of the sensitivity or resistance of tumor cells to chemotherapeutic drugs in vitro as well as MDR1 mRNA expression, tumor origin (primary or secondary), cytogenetic abnormalities, and aberrant immunophenotype was developed. They collected 53 patient sample of acute myeloid leukemia admitted to the Novosibirsk Hematology Center from January 1, 2014, to December 31, 2018 for this purposed and after analysis concluded that the risk stratification of AML patients in accordance with the developed prognostic scale correlates well with the response to therapy and represents an independent predictive factor for the overall survival of patients with newly diagnosed AML.

Overall, the article is of clinical significance, although the number of patients size is very limited. I recommend some minor corrections needed to be done before further process.  

I suggest the following:

1- The English language of the article needs to be improved significantly. Therefore, authors should consult native English speaker for editing the manuscript.

2- In abstract section line 5 rather than word “today” use “now a days”.

3- In abstract section the recruited number of patients along with little description of the patients should be added

4- In materials & methods section: In the patients’ recruitment section, the inclusion and exclusion criteria should be mentioned for patients’ recruitment. Also, the gender information of the patients should be added among the 53 patients in methods section rather than in results section.  

5- The patient size is limited to only 53 while the duration of the collection of samples spread in four years’ time from 2014 to 2018. Please mention the reason why it took them to collect this small number of samples, was the inclusion criteria was too specific or selective, describe in detail.

6- In conclusion the researchers should add a line or two describing the limitations of the study such as low number of samples and in future needs to add more cohort samples for better statistical power.

The English language of the manuscript needed to be improved. 

Author Response

Dear Reviewer 1,

We are very grateful to you for your valuable comments and suggestions that helped us to improve the manuscript. We revised and modified the manuscript according to your comments (revised parts are marked by red).

1- The English language of the article needs to be improved significantly. Therefore, authors should consult native English speaker for editing the manuscript.

According to your recommendation regarding English language and style, the manuscript was copyedited by An AI-based English editing and proofreading web-service TRINKA (https://www.trinka.ai).

2- In abstract section line 5 rather than word “today” use “now a days”.

Corrected. Please, see lines 15-16.

3- In abstract section the recruited number of patients along with little description of the patients should be added.

The information concerning the number and brief characteristics of patients recruited in the study was added in the Abstract. Please, see lines 23-24.

4- In materials & methods section: In the patients’ recruitment section, the inclusion and exclusion criteria should be mentioned for patients’ recruitment. Also, the gender information of the patients should be added among the 53 patients in methods section rather than in results section.

The inclusion and exclusion criteria for patients’ recruitment were added in the Materials and methods section. Please, see lines 113-118.

The information concerning average age and gender distribution of patients was transferred from the Results section to the Materials and Methods section. Please, see lines 99-101.

5- The patient size is limited to only 53 while the duration of the collection of samples spread in four years’ time from 2014 to 2018. Please mention the reason why it took them to collect this small number of samples, was the inclusion criteria was too specific or selective, describe in detail.

We fully agree that the number of patients enrolled in the study is very limited. The reason for this is quite strict criteria for inclusion and exclusion of patients in the study (Inclusion criteria: patients diagnosed with AML on the base of clinical, morphological, and genetic markers according to the standard diagnostic protocols for AML. Exclusion criteria: patients presenting with hematological disorders (e.g., anemia) because of other non-malignant causes, patients with HIV, hepatitis B and C, and tuberculosis). This information was added in the Materials and methods section. Please, see lines 113-118.

During the study, patients with AML were assigned additionally into two cohorts: patients received intensive induction chemotherapy with anthracycline-based regimens and patients received non-intensive chemotherapy with low-dose cytarabine because of their comorbidity and complicated clinical and hematological anamnesis. Thus, patients who, for objective reasons, received other treatment regimens (for example, 6-mercapropurine) were not included in the study, despite their eligibility with the inclusion criteria.

Besides this, some patients who met the inclusion criteria and were initially enrolled in the study did not complete the course of therapy because of the development of severe complications required a shift in treatment regimen, poor compliance or death and therefore were excluded from the analysis.

However, despite the relatively small number of patients enrolled, we found moderate and strong statistically significant correlations between the studied variables, which may indicate the correctness of our approach.

6- In conclusion the researchers should add a line or two describing the limitations of the study such as low number of samples and in future needs to add more cohort samples for better statistical power.

We added the paragraph describing the limitations of the study after the Conclusions section. Please, see lines 588-590.

Comments on the Quality of English Language:

The English language of the manuscript needed to be improved.

Corrected. The manuscript was copyedited by An AI-based English editing and proofreading web-service TRINKA (https://www.trinka.ai).

Reviewer 2 Report

This work aims at establish an initial assessment of the sensitivity of tumor cells to chemotherapeutic agents used in the induction therapy of leukemia to predict the patient's response to antitumor therapy. The prognostic scale proposed by the authors for risk stratification of patients with AML is based on the detection of the sensitivity or resistance of tumor cells to chemotherapeutic drugs in vitro, MDR1 mRNA expression, tumor origin (primary or secondary), cytogenetic abnormalities, and aberrant immunophenotype.

Overall, this work provides a laborious way to re-stratify patients, but it does not seem so efficacious nor innovative. In particular, no survival curve was presented with the new proposed classification to define the robustness of this new classification; moreover, pGp levels, MDR1 and response in vitro to drug were parameters already used by authors in precedent papers, thus reducing the novelty.

Other concerns:

- Among primary chemoresistant patients, authors stated that males predominated with 32.1% vs 28.3%, that is not a relevant difference, not corroborated with a statistical testing.

- The sentence “Thus, it can be assumed that some patients classified as a favorable risk category were initially resistant to induction chemotherapy (compare data in Tables S7, S8 and S9).” Is true only for comparison of S9 vs S7. Conversely, in table S8, the favorable patients were 18.9%, that could be putatively all included in the 30.2% of patients who reached CR.  

- multidrug resistance (MDR) phenotype of tumor cells (responsiveness of tumor cells to cytotoxic drugs in vitro, MDR1 mRNA/P-gp expression in tumor cells):

- how the authors classify the responsiveness to cytotoxic drugs in vitro? In the methods section is reported the IC50 calculation and the doses used, but there are no dose-response curves showed, IC50 value or other data that clarify this classification.

Author Response

Dear Reviewer 2,

We are very grateful to you for your valuable comments and suggestions that helped us to improve the manuscript. We revised and modified the manuscript according to your comments (revised parts are marked by red).

Overall, this work provides a laborious way to re-stratify patients, but it does not seem so efficacious nor innovative. In particular, no survival curve was presented with the new proposed classification to define the robustness of this new classification; moreover, pGp levels, MDR1 and response in vitro to drug were parameters already used by authors in precedent papers, thus reducing the novelty.

Survival of patients who were assigned to the low risk group were significantly higher, than in patients who were assigned to a high and intermediate risk groups according to our prognostic scale (p=0.02535), which indicates high predictive value and reliability of our classification. Overall survival of patients of the low risk group was 58% (6-month survival), 58% (1-year survival) and 46 % (3-year survival) with a medium survival rate 452 days. Overall survival of patients classified as intermediate risk group was follows: 6-month survival 53%, 1-year survival 28%, and 3-year survival 12% with medium survival rate 239 days. Patients assigned to the high risk group demonstrated 6-month survival 30%, 1-year survival 25%, and 3-year survival 10% with medium survival rate 79 days. We included all survival data (please, see lines 393-403) and survival curves (please, see Figure 1) in the Results section.

We fully agree that response to drugs in vitro, P-glycoprotein (P-gp) and MDR1 mRNA expression levels in tumor cells were already used for therapy response and survival prediction in patient with various cancers. However, in most studies, these parameters, reflecting the multidrug resistance phenotype, were taken into account separately. For example, in the recent studies of Zu et al. and Angori et al. drug sensitivity testing in vitro were performed for gastric and renal cancer [1,2]; in the study of Sawazaki et al. P-gp expression were assessed as predictive factors for bladder cancer recurrence [3]; in turn, Kadioglu et al. demonstrated that low ABCB1 mRNA expression correlates with significantly longer survival in thymoma, ovarian and kidney cancers [4]. As for hematological disorders, in particular AML, a large number of works and reviews show clinical relevance of drug transporter proteins in AML patients [5–10]. We added the lines concerning the evidence of drug resistance phenotype detection for leukemia patients in the Discussion section (please, see lines 507-510).

Unfortunately, we did not find works where these parameters (drug responsiveness in vitro, MDR1 mRNA and P-gp expression levels in tumor cells) were joined into one prognostic system. Our study represents a combined approach: enhancement of predictive power of established stratification scales was achieved by integating a number of parameters reflecting the multidrug resistance phenotype in tumor cells. We also compared the known risk stratification scales and our classification in terms of survival and response to therapy in patients with AML. Correlation analysis and Cox regression showed a higher predictive power of our scale compared to risk stratification by genetics for patients received intensive induction chemotherapy with anthracycline-based regimens (please, see Tables 5-7).

It should be noted, that attemts to improve risk and prognosis stratification in AML using new high-cost technologies [11–13] are currently being made. Our approach based on the evaluation of the drug responsiveness of the leukemic cells in vitro does not require expensive equipment and can be useful for a preliminary assessment of the response of AML patients to a number of components of standard induction chemotherapy, which may be of particular value and applicability in low- and middle-income countries limited in full cytogenetic and molecular information at diagnosis. Moreover, current strategies for assessment of multidrug resistance phenotype such as gene expression markers and mutational status of leukemic cells as well as techniques based on genetic profiling are time consuming. Our approach allows to determine a treatment strategy within 5 days of diagnosis. Thus, once drug resistance can be detected in patients' tumor cells, the relevant approaches may be taken to overcome it.

Other concerns:

- Among primary chemoresistant patients, authors stated that males predominated with 32.1% vs 28.3%, that is not a relevant difference, not corroborated with a statistical testing.

Corrected. We remoived the statement about male predominance among the primary chemoresistant patients. Please, see lines 264-265.

- The sentence “Thus, it can be assumed that some patients classified as a favorable risk category were initially resistant to induction chemotherapy (compare data in Tables S7, S8 and S9).” Is true only for comparison of S9 vs S7. Conversely, in table S8, the favorable patients were 18.9%, that could be putatively all included in the 30.2% of patients who reached CR.

Corrected. We removed the reference to Table S8 in the context of comparing gender distribution, risk stratification and response to therapy. Please, see line 268.

- multidrug resistance (MDR) phenotype of tumor cells (responsiveness of tumor cells to cytotoxic drugs in vitro, MDR1 mRNA/P-gp expression in tumor cells): how the authors classify the responsiveness to cytotoxic drugs in vitro? In the methods section is reported the IC50 calculation and the doses used, but there are no dose-response curves showed, IC50 value or other data that clarify this classification.

The present study is the logical continuation of our previous works in which IC50 values used for drug sensitivity asssement were described in detail [14,15], so we did not duplicate information in the present article in order to avoid similarity and manuscript overload.

The IC50 values were obtained by fitting dose-response curves according to the following exponential decay equation using non-linear regression: y = y0 + A1e −(x − x0)/t1, where x is the concentration of the substance and y = 50% (a representative curve is shown in the Figure below).

Figure. Dependence of the number of live cells on the daunorubicin concentration according to the WST-1 test. Conditions: 105 cells per well; concentrations of daunorubicin: 0, 0.05, 0.1, 0.2, 0.4, 0.6, 1 and 2 μM; time of incubation with daunorubicin 72 h; time of exposure to WST-1 solution 3 h. Calculating in Excel and plotting in OriginPro, the IC50 value for daunorubicin in this case was 1.3±0.3 µM/L.

At the initial stages of the study, we evaluated generaly the kinetics of cell death at various concentrations of cytotoxic drugs (daunorubicin and cytarabine) in AML patients with 100% blast cells in the bone marrow, without mutations and received remission after 2 courses of chemotherapy. The final sensitivity range included approximate dose intervals at IC50, IC75 and IC 100. Then, we scaled the drug sensitivity of leukemia cells for subsequent analysis based on IC50 values: scale 1 corresponds to a high sensitivity of tumor cells, and scales 2 and 3 correspond to a moderate and low sensitivity of leukemic cells, respectively (Table).

Table . Scales for the distribution of AML patients according to the sensitivity of their tumor cells to chemotherapeutic drugs.

Chemotherapeutic drug

IC50 values

1 – high sensitivity

2 – moderate sensitivity

3 – low sensitivity (resistance)

Daunorubicin, µM

0-0.25

0.25-0.5

>0.5

Cytarabine, µM

0-1.5

1.5-8

>8

In this way, all leukemia patients enrolled in the study were divided into groups according to the sensitivity of their tumor cells to daunorubicin and cytarabine based on IC50 values.

Table deciphering the scales for the distribution of AML patients according to the sensitivity of their tumor cells to chemotherapeutic drugs was added to the Materials and Methods section. Please, see lines 149-152 and Table 1.

  1. Zu, M.; Hao, X.; Ning, J.; Zhou, X.; Gong, Y.; Lang, Y.; Xu, W.; Zhang, J.; Ding, S. Patient-derived organoid culture of gastric cancer for disease modeling and drug sensitivity testing. Biomed. Pharmacother. 2023, 163, 114751, doi:10.1016/J.BIOPHA.2023.114751.
  2. Angori, S.; Banaei-Esfahani, A.; Mühlbauer, K.; Bolck, H.A.; Kahraman, A.; Karakulak, T.; Poyet, C.; Feodoroff, M.; Potdar, S.; Kallioniemi, O.; et al. Ex Vivo Drug Testing in Patient-derived Papillary Renal Cancer Cells Reveals EGFR and the BCL2 Family as Therapeutic Targets. Eur. Urol. Focus 2023, doi:10.1016/J.EUF.2023.03.005.
  3. Sawazaki, H.; Ito, K.; Asano, T.; Kuroda, K.; Horiguchi, A.; Tsuda, H.; Asano, T. Expressions of P-Glycoprotein, Multidrug Resistance Protein 1 and Annexin A2 as Predictive Factors for Intravesical Recurrence of Bladder Cancer after the Initial Transurethral Resection and Immediate Single Intravesical Instillation of Adriamycin. Asian Pac. J. Cancer Prev. 2021, 22, 1459–1466, doi:10.31557/APJCP.2021.22.5.1459.
  4. Kadioglu, O.; Saeed, M.E.M.; Munder, M.; Spuller, A.; Greten, H.J.; Efferth, T. Effect of ABC transporter expression and mutational status on survival rates of cancer patients. Biomed. Pharmacother. 2020, 131, doi:10.1016/J.BIOPHA.2020.110718.
  5. Vasconcelos, F.C.; de Souza, P.S.; Hancio, T.; de Faria, F.C.C.; Maia, R.C. Update on drug transporter proteins in acute myeloid leukemia: Pathological implication and clinical setting. Crit. Rev. Oncol. Hematol. 2021, 160, doi:10.1016/J.CRITREVONC.2021.103281.
  6. da Silveira Júnior, L.S.; Soares, V. de L.; Jardim da Silva, A.S.; Gil, E.A.; Pereira de Araújo, M. das G.; Merces Gonçalves, C.A.; Paiva, A. de S.; Moura de Oliveira, T.M.; Oliveira, G.H. de M.; Kramer Cavacanti e Silva, D.G.; et al. P-glycoprotein and multidrug resistance-associated protein-1 expression in acute myeloid leukemia: Biological and prognosis implications. Int. J. Lab. Hematol. 2020, 42, 594–603, doi:10.1111/IJLH.13241.
  7. Guo, X.; Shi, P.; Chen, F.; Zha, J.; Liu, B.; Li, R.; Dong, H.; Zheng, H.; Xu, B. Low MDR1 and BAALC expression identifies a new subgroup of intermediate cytogenetic risk acute myeloid leukemia with a favorable outcome. Blood Cells. Mol. Dis. 2014, 53, 144–148, doi:10.1016/J.BCMD.2014.05.001.
  8. Yazdandoust, E.; Sadeghian, M.H.; Shams, S.F.; Ayatollahi, H.; Saadatpour, Y.; Siyadat, P.; Sheikhi, M.; Afzalaghaee, M. Evaluation of FLT3-ITD Mutations and MDR1 Gene Expression in AML Patients. Iran. J. Pathol. 2022, 17, 419–426, doi:10.30699/IJP.2022.543485.2776.
  9. Rodríguez-Macías, G.; Briz, O.; Cives-Losada, C.; Chillón, M.C.; Martínez-Laperche, C.; Martínez-Arranz, I.; Buño, I.; González-Díaz, M.; Díez-Martín, J.L.; Marin, J.J.G.; et al. Role of Intracellular Drug Disposition in the Response of Acute Myeloid Leukemia to Cytarabine and Idarubicin Induction Chemotherapy. Cancers (Basel). 2023, 15, 3145, doi:10.3390/CANCERS15123145.
  10. Kaltoum, A.B.O.; Sellama, N.; Hind, D.; Yaya, K.; Mouna, L.; Asma, Q. MDR1 gene polymorphisms and acute myeloid leukemia AML susceptibility in A Moroccan adult population: A case-control study and meta-analysis. Curr. Res. Transl. Med. 2020, 68, 29–35, doi:10.1016/J.RETRAM.2019.06.001.
  11. Wang, Q. Five cuprotosis-related lncRNA signatures for prognosis prediction in acute myeloid leukaemia. https://doi.org/10.1080/16078454.2023.2231737 2023, 28, doi:10.1080/16078454.2023.2231737.
  12. Liang, X.; Li, C.; Fan, M.; Zhang, W.; Liu, L.; Zhou, J.; Hu, L.; Zhai, Z. Immune-related lncRNAs pairs prognostic score model for prediction of survival in acute myeloid leukemia patients. Clin. Exp. Med. 2023, doi:10.1007/S10238-023-01085-2.
  13. Liu, Y.; Cao, Y.; Yang, X.; Chen, H.; Yang, H.; Liu, Y.; Gu, W. High expression of miR-107 and miR-17 predicts poor prognosis and guides treatment selection in acute myeloid leukemia. Transl. Cancer Res. 2023, 12, 913–927, doi:10.21037/TCR-22-2484.
  14. Kolesnikova, M.; Sen’Kova, A.; Tairova, S.; Ovchinnikov, V.; Pospelova, T.; Zenkova, M. Clinical and Prognostic Significance of Cell Sensitivity to Chemotherapy Detected in vitro on Treatment Response and Survival of Leukemia Patients. J. Pers. Med. 2019, 9, doi:10.3390/JPM9020024.
  15. Kolesnikova, M.A.; Sen’kova, A. V.; Pospelova, T.I.; Zenkova, M.A. Drug responsiveness of leukemic cells detected in vitro at diagnosis correlates with therapy response and survival in patients with acute myeloid leukemia. Cancer Rep. 2021, 4, doi:10.1002/cnr2.1362.

Round 2

Reviewer 2 Report

The authors improve the manuscript and the significance of their results, adding a schema representing the sensitivity of patients that clarify their novel classification, and adding the survival curves. 

The Kaplan–Meier survival curves usually require long rank test that gives an overall p-value, instead of comparing single curves. Please modify your statistical analysis accordingly.  

Author Response

Dear Reviewer 2,

We are very grateful to you for high appraisal of our manuscript!

The Kaplan–Meier survival curves usually require long rank test that gives an overall p-value, instead of comparing single curves. Please modify your statistical analysis accordingly.

We revised and modified the survival data according to your comments (please, see Figure 1). Overall p values are represented in Figure 1.
